# Luminal STIM1 Mutants that Cause Tubular Aggregate Myopathy Promote Autophagic Processes

**DOI:** 10.3390/ijms21124410

**Published:** 2020-06-21

**Authors:** Matthias Sallinger, Adéla Tiffner, Tony Schmidt, Daniel Bonhenry, Linda Waldherr, Irene Frischauf, Victoria Lunz, Isabella Derler, Romana Schober, Rainer Schindl

**Affiliations:** 1Institute of Biophysics, JKU Life Science Center, Johannes Kepler University Linz, A-4020 Linz, Austria; matthias.sallinger@jku.at (M.S.); adela.tiffner@jku.at (A.T.); irene.frischauf@jku.at (I.F.); vici.lunz@gmail.com (V.L.); isabella.derler@jku.at (I.D.); 2Gottfried Schatz Research Center, Medical University of Graz, A-8010 Graz, Austria; tony.schmidt@medunigraz.at (T.S.); linda.waldherr@medunigraz.at (L.W.); 3Center for Nanobiology and Structural Biology, Institute of Microbiology, Academy of Sciences of the Czech Republic, CZ-373 33 Nove Hrady, Czech Republic; bonhenry@nh.cas.cz; 4BioTechMed-Graz, A-8010 Graz, Austria

**Keywords:** STIM, Orai, Ca^2+^, SOCE, MITF, TFEB, hydrophobic pocket, EF-hand, tubular aggregate myopathy

## Abstract

Stromal interaction molecule 1 (STIM1) is a ubiquitously expressed Ca^2+^ sensor protein that induces permeation of Orai Ca^2+^ channels upon endoplasmic reticulum Ca^2+^-store depletion. A drop in luminal Ca^2+^ causes partial unfolding of the N-terminal STIM1 domains and thus initial STIM1 activation. We compared the STIM1 structure upon Ca^2+^ depletion from our molecular dynamics (MD) simulations with a recent 2D NMR structure. Simulation- and structure-based results showed unfolding of two α-helices in the canonical and in the non-canonical EF-hand. Further, we structurally and functionally evaluated mutations in the non-canonical EF-hand that have been shown to cause tubular aggregate myopathy. We found these mutations to cause full constitutive activation of Ca^2+^-release-activated Ca^2+^ currents (I_CRAC_) and to promote autophagic processes. Specifically, heterologously expressed STIM1 mutations in the non-canonical EF-hand promoted translocation of the autophagy transcription factors microphthalmia-associated transcription factor (MITF) and transcription factor EB (TFEB) into the nucleus. These STIM1 mutations additionally stimulated an enhanced production of autophagosomes. In summary, mutations in STIM1 that cause structural unfolding promoted Ca^2+^ down-stream activation of autophagic processes.

## 1. Introduction

Ca^2+^ entry via store-operated Ca^2+^ channels (SOCs) provides an essential signaling mechanism for cellular immune responses and is required to control the filling state of the endoplasmic reticulum (ER) in virtually every cell type [1,2,3,4]. Activation of this sustained SOC entry (SOCE) is provided on a molecular basis by the ER Ca^2+^-sensing proteins stromal interaction molecule 1 (STIM1) and STIM2 and the plasma membrane (PM) Ca^2+^ channels Orai1, 2, and 3 [5,6,7,8,9]. STIM1 senses luminal Ca^2+^ with a canonical EF-hand and responds to a reduced Ca^2+^ concentration within the ER [10,11]. Resting Ca^2+^ concentrations are at around 800 µM, while a release of ER Ca^2+^, physiologically mediated by inositol-3-phosphate or ryanodine receptors, leads to a drop in ER Ca^2+^ concentrations to 100–400 µM [12,13]. As a consequence, STIM1 loses bound Ca^2+^ within its canonical EF-hand [12,14]. Functionally, this Ca^2+^-binding is important to keep the luminal STIM1 structure folded [15,16,17]. Ca^2+^-free conditions partially destabilize the luminal STIM1 and result preferentially in STIM1 dimers [11,15]. Even larger aggregates arise from C-terminal STIM1 multimerization [18,19]. However, the STIM C-terminus is not only responsible for STIM clustering, but also extends and binds directly to the Orai1 channel [20,21,22,23,24,25,26,27] at specifically localized ER-PM junctions, where the membranes are only 10 to 17 nm apart [12,18,28]. The STIM/Orai complexes are stable over long time periods to mediate Ca^2+^ signaling for several minutes up to hours [29]. Ca^2+^ influx initiates local signaling cascades, such as the well-characterized activation of the nuclear factor of activated T-cells (NFAT) [4,30,31,32,33]. 

Thus, luminal STIM1 Ca^2+^-dependent unfolding is a key prerequisite for the activation of a precisely controlled SOCE signaling cascade. The luminal structures of STIM1 and STIM2 have been determined and show a canonical and a non-canonical EF-hand and a sterile alpha motif (SAM) domain (Figure 1A) [11,17,34]. In the Ca^2+^-bound state of STIM, all three domains are tightly packed together [11,17]. NMR structures determined a hydrophobic pocket formed by amino acids in the canonical, the non-canonical EF-hand, and the SAM domain [11,17,34]. These hydrophobic pocket residues in human STIM1 are V68, I71, H72, M75, L92, L96, and K104 within the canonical EF-hand, F108, H109, I115, and L120 in the non-canonical EF-hand, and the residues L195 and L199 in the SAM domain. It is of great interest that mutations in several of these hydrophobic pocket residues within the canonical (H72Q and L96V) and non-canonical (F108I, H109N, H109R, and I115F) EF-hands result in tubular aggregate myopathy [35,36]. In line with this finding, mutations that interfere with the hydrophobic pocket within the canonical EF-hand (H72R or H72Q, L92P, L96V), the non-canonical EF-hand (F108I, F108D-G110D, H109N or H109R and I115F), and the SAM domain (L195R) result in aggregation, constitutively active Ca^2+^ entry, or NFAT activation [11,15,35,36]. 

In this research, we investigate whether these non-canonical EF-hand mutations stimulate Ca^2+^-mediated signaling pathways in autophagy and examined transcriptional processes and autophagosome formation. We therefore focused our attention on the calcium-dependent transcription factor EB (TFEB) and its close homolog, the microphthalmia-associated transcription factor (MITF). Both transcription factors are part of the MIT/TFE family and are master regulators of lysosomal biogenesis, autophagy, mitophagy, and stress response [37,38,39,40]. Activation of TFEB is mediated by the Ca^2+^/Calmodulin-dependent phosphatase Calcineurin (CaN) [41]. Dephosphorylation of TFEB controls the translocation of this transcription factor to the nucleus [41,42]. This CaN-dependent phosphorylation is exquisitely similar to STIM1/Orai1-mediated translocation of NFAT [32,33], and previous research has already associated SOCE with TFEB activation in pancreatic acinar cells [43]. The rise in Ca^2+^ concentration responsible for TFEB activation originates either from SOCE or from lysosomal store depletion [41,43,44].

Specifically, in the resting state, TFEB and MITF are phosphorylated at multiple sites, which is a prerequisite for 14-3-3 binding [45]. The 14-3-3 protein is responsible for cytosolic sequestration, possibly due to the masking of a nuclear localization signal (NLS) [42]. Hence, CaN dephosphorylates TFEB, which allows its nuclear translocation and activates expression of autophagy genes [41]. TFEB activity can be linked specifically to autophagosome promotion, as autophagosome formation is reduced upon silencing of CaN [41,43,46]. This identifies CaN as a regulator of the autophagic pathway [41]. However, autophagosome formation can also take place in a transcription-independent manner, hence in a TFEB-independent manner, as lysosomal Ca^2+^-store depletion via transient receptor potential mucolipin 1 (TRPML1) is sufficient to induce autophagosome biogenesis [47].

Here, we link tubular aggregate myopathy (TAM) mutations in STIM1 to their role in destabilizing the hydrophobic pocket and in promoting autophagic pathways. Specifically, we found that mutations of hydrophobic residues that form the non-canonical EF-hand result in constitutively active Ca^2+^ influx. In addition, constitutively active STIM1 mutants of both the non-canonical and the canonical EF-hand induced autophagic processes, including activation of the calcium-dependent transcription factors MITF and TFEB. Furthermore, we determined enhanced autophagosome formation upon overexpression of constitutively active STIM mutants.

## 2. Results

### 2.1. The Hydrophobic Pocket of STIM1 Unfolds upon Ca^2+^ Depletion or Due to a Tubular Aggregate Myopathy Mutation

We initially used molecular dynamics (MD) simulations to investigate how the resting-state luminal STIM1 (Figure 1A,B) rearranges to reach its Ca^2+^-depleted conformation (Figure 1C). We visualized the helices of the NMR structure of STIM1 in its Ca^2+^-bound resting state as cylinders (Figure 1A–C) [11]: the canonical (helices 1 and 2) and the non-canonical (helices 3 and 4) EF-hands are connected via the short helix 5 to the SAM domain (helices 6 to 10) (Figure 1A,B and [11]). The helix nomenclature is in accordance with that in previous research [11]. MD simulations revealed that, in the Ca^2+^-depleted state, only helices 1 and 4 of the canonical and non-canonical EF-hands, respectively, are maintained (Figure 1C). Helices 2 and 3 exhibit a large unfolded loop (Figure 1C, lower panel). In agreement with our MD simulations, helices 2 and 3 remained unassigned in the 2D NMR, which suggests an unfolded secondary structure [17]. Furthermore, in line with our results, helices 1 and 4 are maintained, and the SAM domain (helices 6 to 10) together with the connecting helix 5 remain folded both in 2D NMR and in MD simulations [17].

We investigated STIM1 residues that are associated with tubular aggregate myopathy [35,36] when mutated (F108I, H109N/R, I115F; Figure 2A, lower panel). These residues are located within the non-canonical EF-hand of STIM1 and form part of the hydrophobic pocket (Figure 2A, upper panel). The hydrophobic pocket residues are tightly packed together in the center of the folded STIM1 structure and keep the resting STIM1 folded (Figure 2B) [11,15]. The greatest reorientation in hydrophobic canonical EF-hand residues upon transition from the resting to the active state is visible for L92 and L96. Both residues are substantially separated from the core of the STIM1 structure under Ca^2+^-free conditions (Figure 2C). Previous results have already highlighted the importance of these two leucines, and their mutation results in constitutive STIM1 activation [15]. Two non-canonical EF-hand residues, F108 and H109, are also detached from the remaining core of the hydrophobic pocket (Figure 2C).

An F108I mutation, found in patients with tubular aggregate myopathy, causes constitutive Ca^2+^ signaling [15], so we analyzed it further in MD simulations. STIM1-F108I mutation yielded unfolding of helix 3 (Appendix A), which includes hydrophobic pocket residues K104, F108, and H109 (Figure 2D). In the simulations, these residues are separated from the rest of the hydrophobic pocket. All other luminal helices remain intact due to the presence of stabilizing Ca^2+^ ions (Appendix A). This accords with the finding that purified STIM1-F108I fragments unfold earlier in temperature-dependent measurements [15]. Next, we quantified to which extent each hydrophobic pocket residue is unfolded in (i) the Ca^2+^-free STIM1 structure and (ii) the STIM1-F108I mutation. We evaluated all hydrophobic pocket residues of the canonical and non-canonical EF-hands for their relative mean distance to the center of the EF-hands in the last 100 ns of the MD simulations (Figure 2E). As expected from the visualization of the Ca^2+^-free, unfolded STIM1 structure, L92 and L96 showed a high degree of unfolding, while the rest of the canonical and non-canonical EF-hands was only partially unfolded (Figure 2E; middle) [15]. In the STIM1-F108I simulation, K104 and I108 exhibited the highest degree of unfolding. The amount of unfolding of all hydrophobic residues was in general higher for the Ca^2+^-free STIM1 than for STIM1-F108I.

### 2.2. Puncta Formation and Constitutive I_CRAC_ Currents Are Induced by STIM1 Hydrophobic Pocket TAM Mutations

We additionally investigated STIM1 hydrophobic-pocket TAM mutants at the cellular level and selected single-point mutations in the non-canonical EF-hand (F108I, H109N, H109R, and I115F) for live-cell experiments. First, we evaluated the localization of YFP-tagged STIM1 wild type and mutants upon overexpression in HEK cells. Wild type STIM1 exhibited tubular localization in the resting state [48] (Figure 3A, upper panel). ER Ca^2+^-store depletion was induced by treatment with thapsigargin (TG, 1 µM, 5 min), an inhibitor of Sarcoplasmic/endoplasmic reticulum calcium ATPase (SERCA) pumps. Consequently, wild type STIM1 rearranged and exhibited puncta formation (Figure 3A, lower panel). In contrast, STIM1 non-canonical EF-hand mutants (F108I, H109N, H109R, and I115F) already exhibited pre-clustered localization under resting conditions, similarly to STIM1 wild type upon ER store depletion (Figure 3B). Addition of TG did not further increase clustering of these STIM1 mutants, which indicates that these mutations switch STIM1 into a constitutively active state. These experiments are very well in line with our recent MD simulations on STIM1-F108I that loses dynamic Ca^2+^ interactions of the canonical EF-hand over time [15]. 

To evaluate whether these STIM1 hydrophobic pocket mutants are able to fully activate endogenous store-operated channels, we performed whole-cell patch-clamp experiments in rat basophilic leukemia (RBL) cells. This cell type has been very well described for large Ca^2+^-release-activated Ca^2+^ currents (I_CRAC_) [49,50]. Store depletion for the activation of Ca^2+^ signaling of STIM1 wild type cells was induced by EGTA in the patch pipette. EGTA complexes Ca^2+^ ions with high affinity and results in delayed passive ER Ca^2+^-store depletion. Overexpression of wild type STIM1, therefore, resulted in delayed activation and reached a maximum I_CRAC_ after ~200 s (Figure 3C). In contrast, each of the overexpressed STIM1 non-canonical EF-hand mutants (F108I, H109N, H109, and I115F) exhibited robust constitutively active I_CRAC_ immediately after whole-cell break-in (Figure 3C,D). The current/voltage relationship of maximum currents revealed a similar inward-rectification of I_CRAC_ generated by STIM1 wild type and the mutants (Figure 3E). 

I_CRAC_ is strictly required to stimulate nuclear translocation and thus activate NFAT [31,32]. We have previously overexpressed STIM1-F108I, H109N, H109R, and I115F mutants individually and determined a preferentially higher nuclear localization of NFAT under resting ER Ca^2+^ conditions than for wild type STIM1 [15,33]. This NFAT translocation screen has offered an effective approach to identifying constitutively active STIM1 mutants [15]. Increased intracellular Ca^2+^ concentration due to I_CRAC_ is directly linked to NFAT-regulated transcriptional activity which is controlled by the Ca^2+^- and Calmodulin-dependent phosphatase Calcineurin [33,51,52].

### 2.3. Puncta Formation and Constitutive I_CRAC_ Currents Are Induced by STIM1 Hydrophobic Pocket TAM Mutations

In a remarkably similar activation mechanism, the Ca^2+^/Calmodulin/Calcineurin signaling pathway dephosphorylates also TFEB, triggering nuclear translocation [41]. We investigated whether heterologous overexpression of YFP-tagged STIM1 F108I, H109N, H109R, or I115F mutants in HEK cells results in preferential nuclear trans-localization of CFP-tagged TFEB and MITF one day after co-transfection. In fact, this set of constitutively active STIM1 mutants exhibited a significant increase both in nuclear MITF (40–55% of cells) (Figure 4A,B, Appendix A) and TFEB (50–65% of cells) localization (Figure 4C,D, Appendix A) compared to cells overexpressing STIM1 wild type (MITF: ~25%; TFEB: ~30% of cells). 

Another Ca^2+^-regulated autophagic process is the biogenesis of autophagic vesicles [47]. This Ca^2+^ signaling pathway links the lysosomal Ca^2+^-release channel TRPML1 via activation of Ca^2+^/Calmodulin-dependent protein kinase kinase ß and AMP-activated protein kinase (AMPK), induction of the Beclin1/VPS34 autophagic complex and generation of phosphatidylinositol 3-phosphate (PI3P) to autophagic vesicle biogenesis [47]. We directly investigated autophagosome biogenesis by determining the number of autophagosomes per cell, using the PI3P reporter GFP-2xFYVE (Figure 5) [41,43,46]. For these experiments, we expanded our set of constitutively active (non-canonical EF-hand) STIM1 mutants to include canonical EF-hand mutants (H72Q, H72R, N80K, and E87Q) in RBL mast cells. We have previously determined that the latter type of mutant exhibits cluster formation under resting cell conditions and induces high constitutive Ca^2+^-entry activity [15]. Overexpression of STIM1 canonical (Figure 5C,D, Appendix A) and non-canonical (Figure 5A,B, Appendix A) EF-hand mutants together with the 2xFYVE-autophagosome reporter (Figure 5B,D) led to significantly more autophagosomes per cell than overexpression of STIM1 wild type. In contrast, mock transfected cells and overexpression of STIM1 wild type yielded a similar low number of autophagosome-positive vesicles under resting cell conditions (Figure 5A,C). 

These experiments show that STIM1 mutations that cause tubular aggregate myopathy result in constitutive activity and have a robust stimulatory effect on autophagosome formation and autophagy transcription factors. These mutations affect the non-canonical EF-hand, but do not directly alter Ca^2+^-ion binding [11,17]. Rather, they destabilize the hydrophobic pocket, which is an essential step in the activation of STIM1.

## 3. Discussion

High-resolution structures in combination with live-cell experiments have demonstrated the importance of luminal STIM1 domains, especially of the canonical and non-canonical EF-hands and the SAM domain, in maintaining a tightly packed protein conformation under resting cell conditions [10,11,17,18,53]. Lowering the Ca^2+^ concentration below resting ER conditions leads to protein destabilization that is induced by unfolding events primarily in the canonical and non-canonical EF-hands [11,17,34]. This partially unfolded STIM1 preferentially forms dimers in the initial step of a precisely coordinated activation cascade that leads to gating of SOCE across the plasma membrane. Our study provides new insights into this structural unfolding process upon Ca^2+^ depletion, especially regarding the non-canonical EF-hand. We combined functional live-cell recordings with MD simulations to reveal the importance of the non-canonical EF-hand in maintaining a hydrophobic pocket. Mutations within this domain have previously been identified in patients suffering from TAM (F108I, H109N, H109R, I115F) [35,36]. In line with previous results, all four TAM mutants investigated resulted in constitutive STIM1 activation and dimerization/oligomerization of STIM1 proteins [15,35,36]. Our MD results showed that all hydrophobic pocket residues are involved in the unfolding process upon Ca^2+^ depletion. This structural rearrangement is most pronounced for the canonical EF-hand residues L92 and L96. Interestingly, a L92P mutation has been identified in cancer patients, and L92V and L96V mutations, which cause constitutive SOC entry, have been found in patients suffering from tubular aggregate myopathy [15,35,36,54]. 

Structural analysis of STIM1 in its resting state has shown a configuration largely similar to that of the ubiquitously expressed Ca^2+^-binding protein Calmodulin [55]. The two EF-hands of STIM1 are wrapped around the SAM domain in a way that closely resembles the configuration of a lobe of Calmodulin in its Ca^2+^-bound state around its target structures [55]. Interestingly, two residues in the non-canonical EF-hand domain show clear functional differences: In a classical EF-hand (e.g., in Calmodulin), the negatively charged side-chains of the STIM1 F108 and G110 positions would contribute to Ca^2+^-binding [11]. Engineered mutations (F108D-G110D) lead to STIM1 dimerization/oligomerization instead [11]. These results clearly highlight that the function of these two residues has adapted to the relatively high Ca^2+^ concentration of the ER under resting conditions. This accords with biochemical experiments that have shown the STIM1-F108I mutant to be less temperature-stable and to have reduced Ca^2+^ affinity [11,15]. Results in this research visualize the unfolding of helix 3 of the non-canonical EF-hand and destabilization of the hydrophobic pocket. Consequently, STIM1-F108I yields maximally activated I_CRAC_ currents, which underlines the importance of the hydrophobic residue in this position. We expect similar structural effects also for other non-canonical EF-hand TAM mutants. As example, the substitution of H109 to more hydrophilic side-chains would no longer favor an orientation of the mutated arginine or asparagine in the hydrophobic pocket and hence would destabilize this pocket. 

Destabilization of the luminal STIM1 domain is a starting point for dimerization, which ultimately leads to activation of SOCE. Our study showed that heterologously expressed STIM1 EF-hand mutants exhibit maximally constitutively active SOCE channels, which causes unbalanced intracellular Ca^2+^ concentrations (see Results section and [15]). Subsequent increase in intracellular Ca^2+^, either induced by SOCE or lysosomal Ca^2+^-store depletion, was shown to activate signaling cascades associated with autophagic processes [41,43]. Here, we have reported increased nuclear localization of the transcription factors MITF and TFEB in cells expressing TAM-associated STIM1 mutants (F108I, H109N, H109R, I115F). Translocation of TFEB is a key signaling event for stimulating lysosomal biogenesis and autophagy [41,43]. Using PI3P reporter 2xFYVE upon co-expression of STIM1 canonical and non-canonical EF-hand domain mutants, we additionally observed enhanced autophagosome biogenesis. Note that gain-of-function STIM1 mutations, including TAM and Stormorken syndrome mutations, result in a broad spectrum of multisystem diseases characterized by muscle weakness, thrombocytopenia, hyposplenism, ichthyosis, dyslexia, moisis, and short stature [54]. Further research is required to properly characterize TAM at the cellular level, and an autophagy phenotype due to constitutively active Ca^2+^ signaling, as determined in this research, has not yet been investigated in TAM patients.

In summary, our results suggest a putative impaired Ca^2+^ homeostasis—induced by constitutively active mutations in the two EF-hands of STIM1—which leads to nuclear translocation of MITF/TFEB and autophagosome formation. Further studies are required to determine (i) whether endogenous SOCE stimulation is sufficient to induce dephosphorylation of the MITF/TFEB transcription factor through the Ca^2+^/Calmodulin-dependent phosphatase Calcineurin and (ii) the potential role of increased cytosolic Ca^2+^ due to SOCE in autophagic vesicle formation.

## 4. Materials and Methods 

### 4.1. Plasmid-DNA

STIM1 DNA constructs (accession number NM_003156), with specific point mutations pre-inserted using the QuikChange site-directed mutagenesis kit (StrataGene, San Diego, CA, USA), were cloned into pECFP-C1 and pEYFP-C1 expression vector systems (Clontech, now Takara, Mountain View, CA, USA). Successful mutation and vector-insertion was confirmed by sequence analysis (Eurofins Genomics, Vienna, Austria).

### 4.2. Transfection

TransFectin Lipid Reagent (BioRad, Vienna, Austria) was used for transient transfection of HEK 293 cells. For each YFP-tagged STIM1 construct or mutant, 1 µg of plasmid-DNA was used. For subcellular localization studies, 4 µg CFP-tagged MITF or 2 µg CFP-tagged TFEB plasmid-DNA was co-transfected. Electrophysiology and 2xFYVE vesicle experiments were performed using rat basophilic leukemia (RBL) cells. RBL cells were electroporated with 7 µg STIM1 wild type or mutant plasmid DNA, and for 2xFYVE experiments, 7 µg 2xFYVE-EGFP was co-transfected. Both cell types were grown for 20–24 h at 37 °C in a humidity-controlled incubator with 5% CO_2_ after transfection/electroporation and regularly tested for mycoplasma contamination. 

### 4.3. Electrophysiology

Electrophysiological experiments were performed with RBL cells 20–24 h after electroporation with 7 µg STIM1 wild type or STIM1 mutants. Whole-cell recording configuration was used at 21–25 °C, with an Ag/AgCl electrode serving as reference electrode. Voltage ramps were applied every 5 s from a holding potential of 0 mV, covering a range of −90 to 90 mV over 1 s. Experiments were performed with bath solution containing 10 mM Ca^2+^, and store-dependent activation was induced using 20 mM EGTA. Standard extracellular solution consisted of (in mM) 145 NaCl, 10 HEPES, 10 CaCl_2_, 10 Glucose, 5 CsCl, and 1 MgCl_2_; pH 7.4. Passive store depletion was induced by an internal pipette solution containing (in mM) 145 Cs methane sulphonate, 20 EGTA, 10 HEPES, 8 NaCl, and 3.5 MgCl_2_; pH 7.2. For liquid junction potential correction, resulting from a Cl^−^-based bath solution and a sulphonate-based pipette solution, +12 mV were applied. All currents were leak-corrected by subtracting the initial voltage ramps obtained shortly after break-in with no visible current activation from the measured currents. All experiments were conducted on at least three different days. 

### 4.4. Confocal Fluorescence Microscopy

Confocal fluorescence microscopy was performed with both RBL and HEK 293 cells to identify subcellular localization of STIM1 wild type and mutants, MITF/TFEB transcription factor family (in HEK 293 cells) and 2xFYVE vesicle formation (in RBL cells). Transfected HEK and electroporated RBL cells were grown on coverslips for 20–24 h at 37 °C in a humidity-controlled incubator with 5% CO_2_. For measurement, the coverslips with the cells were washed and transferred into a solution consisting of 140 mM NaCl, 5 mM KCl, 1 mM MgCl_2_, 2 mM CaCl_2_, 10 mM glucose, and 10 mM HEPES buffer (adjusted to pH 7.4 with NaOH). 

Fluorescence images of the subcellular localization of MITF/TFEB transcription factors were recorded using a QLC100 Real-Time Confocal System (VisiTech Int., Sunderland, UK) connected to two Photometrics CoolSNAPHQ monochrome cameras (Roper Scientific, Planegg, Germany) and a dual-port adapter (dichroic: 505lp; cyan emission filter: 485/30; yellow emission filter: 535/50; Chroma Technology Corp., Olching, Germany). This system was attached to an Axiovert 200M microscope (Zeiss, Oberkochen, Germany) in conjunction with two diode lasers (445 nm, 515 nm) (Visitron Systems, Puchheim, Germany). Visiview 2.1.1 software (Visitron Systems) was used for image acquisition and control of the confocal system. Illumination times for CFP/FRET and YFP images were recorded consecutively with a minimum delay of about 1000 ms. ImageJ was employed for subcellular localization analysis of the transcription factors by means of intensity measurements of the cytosol and nucleus, distinguishing between three different populations with different nucleus/cytosol ratios: inactive (<0.85), homogenous (0.85–1.15), and active (>1.15).

For fluorescence image recording of 2xFYVE vesicle experiments, a CSU-X1 Real-Time Confocal System (Yokogawa Electric Corporation, Vienna, Austria) fitted with two CoolSNAP HQ2 CCD cameras (Photometrics) was used. a dual port adapter (dichroic: 505lp, cyan emission filter: 470/24, yellow emission filter: 535/30, Chroma Technology Corporation) also formed part of the apparatus. All these parts were connected to an Axio Observer Z1 inverted microscope (Zeiss, Oberkochen, Germany) with two diode lasers (445 and 515 nm, Visitron Systems) and placed on a Vision IsoStation anti-vibration table (Newport Corporation, Irvine, CA, USA). 

The VisiView software package (V.2.1.4, Visitron Systems) was used for confocal system control and image generation. The images were created and analyzed on a pixel-to-pixel basis by means of a custom-made software integrated into MATLAB (v7.11.0, The MathWorks, Inc.). Visible 2xFYVE positive vesicle spots for 2xFYVE (mock), STIM1 wild type, and STIM1 mutants in the recorded images were counted and averaged. The same approach, additionally using threshold settings in ImageJ, has been applied for STIM1 cluster quantification.

All experiments were performed on three different days at room temperature, and the resulting data are presented as mean ± SEM (standard error of the mean) for the indicated number of experiments. Unpaired two-sided Student’s *t*-tests for comparison of two groups (performed with Origin Pro 2019) were used to determine significant differences (*p* < 0.01). 

### 4.5. MD Simulation for Analysis of the Unfolding 

All analyses used simulations created in previous research [15]. The mean distances between the centers of the EF-hands and residues belonging to the hydrophobic pocket were calculated using the last 100 ns of simulated STIM1 structures. This was done by subtracting the distances obtained in the folded structure from the distances obtained in the unfolded one for each residue. Finally, all distances were normalized with respect to the greatest distance obtained (L92).

## Figures and Tables

**Figure 1 ijms-21-04410-f001:**
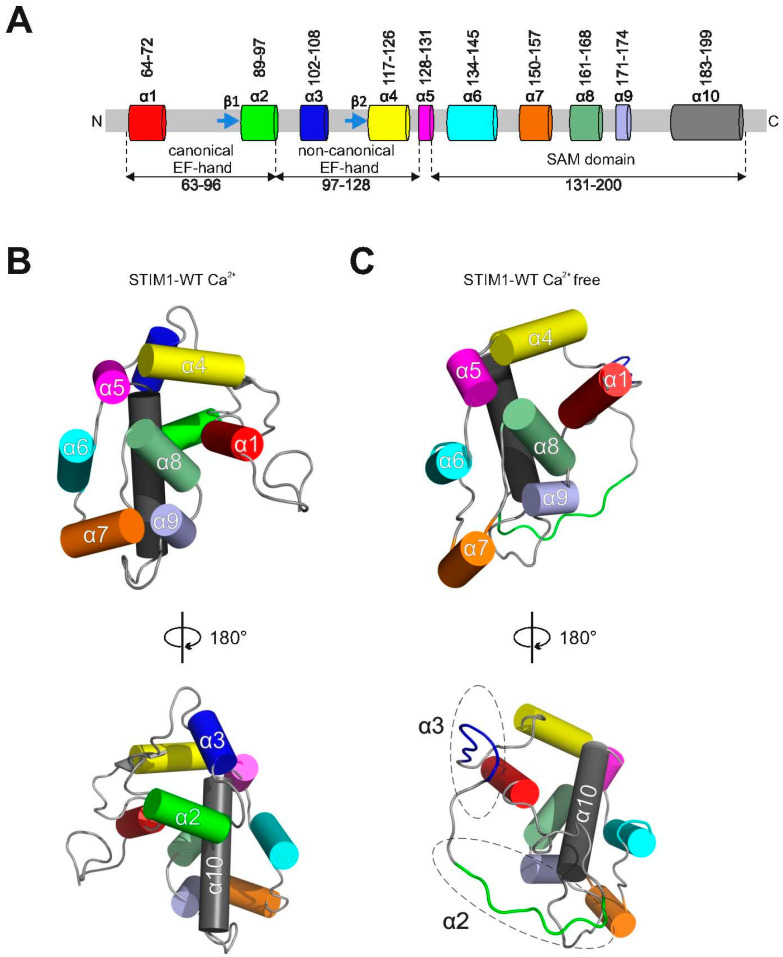
Conformational rearrangement of luminal STIM1 upon Ca^2+^ depletion. (**A**) Schematic representation of the secondary structure of the canonical, non-canonical, and sterile alpha motif (SAM) domains in luminal STIM1 proteins (α-helices are shown as cylinders and β-sheets as arrows). (**B**) Two orientations are shown for the Ca^2+^-bound resting state of STIM1 based on NMR results [11]. The helices (shown as cylinders) are labeled as follows: canonical EF-hand: helices 1 (red) and 2 (green); non-canonical EF-hand: helices 3 (blue) and 4 (yellow); a short connecting helix 5 (purple); and the SAM domain: helices 6 to 10 (turquoise, brown, dark green, light blue, gray). (**C**) Luminal STIM1 structure (from MD simulations) upon Ca^2+^ depletion compared with (**B**) in terms of helix structures. Helices 2 and 3, indicated by dashed circles, are unfolded in MD simulations.

**Figure 2 ijms-21-04410-f002:**
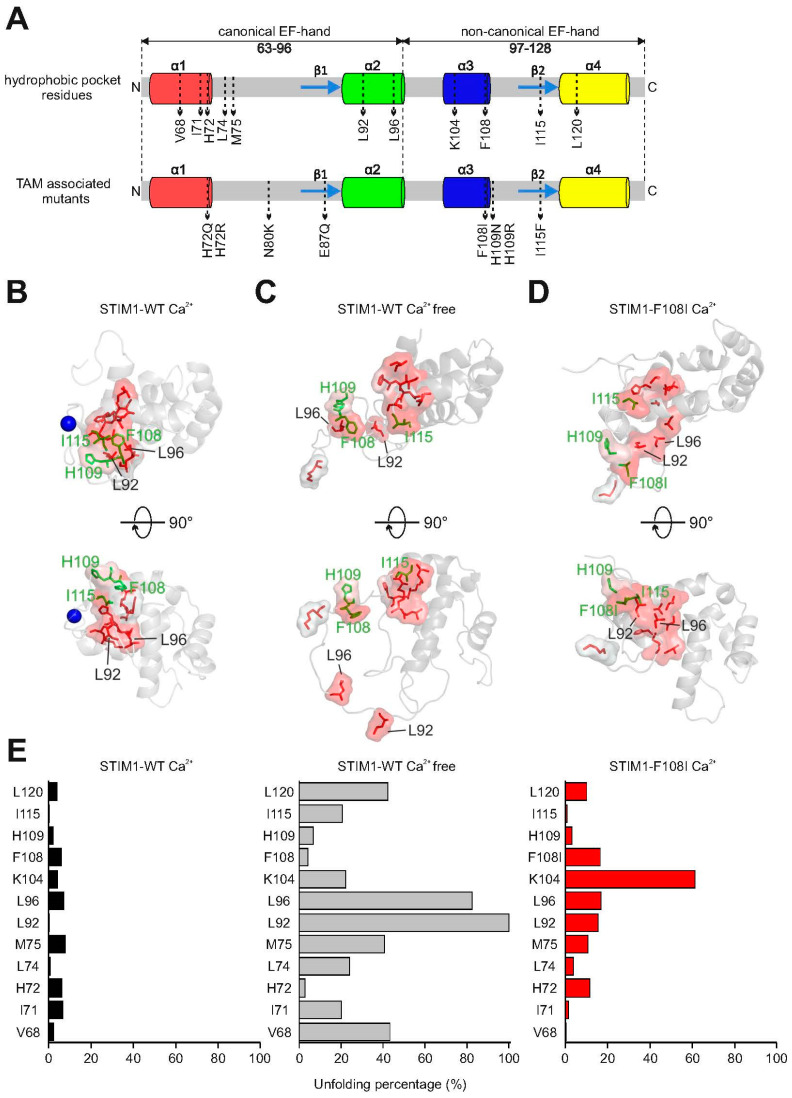
Unfolding of the hydrophobic pocket of STIM1 upon Ca^2+^ depletion. (**A**) Schematic representation of canonical and non-canonical EF-hand structures showing α-helices as cylinders and β-sheets as arrows. In the upper panel, specific amino acids that form the hydrophobic pocket (as described in [11]) are highlighted. The lower panel shows disease-related mutations that are located in the canonical and non-canonical EF-hand domains [35,36]. (**B**–**D**) Hydrophobic pocket residues from panel (**A**) are shown both as red spheres and as stick models in two orientations, (**B**) in the Ca^2+^-bound resting state of STIM1 based on NMR results and a bound Ca^2+^ ion (shown as blue sphere) [11], (**C**) upon Ca^2+^ depletion or (**D**) in a STIM1-F108I mutant (from MD simulations [17]). In addition, hydrophobic residues of the non-canonical EF-hand are shown as green sticks. (**E**) Relative proportion of unfolding (in %) for residues of the canonical and non-canonical EF-hands that contribute to the hydrophobic pocket. The level of unfolding was compared to that of the L92 residue, which showed the highest level of unfolding and was set to 100%.

**Figure 3 ijms-21-04410-f003:**
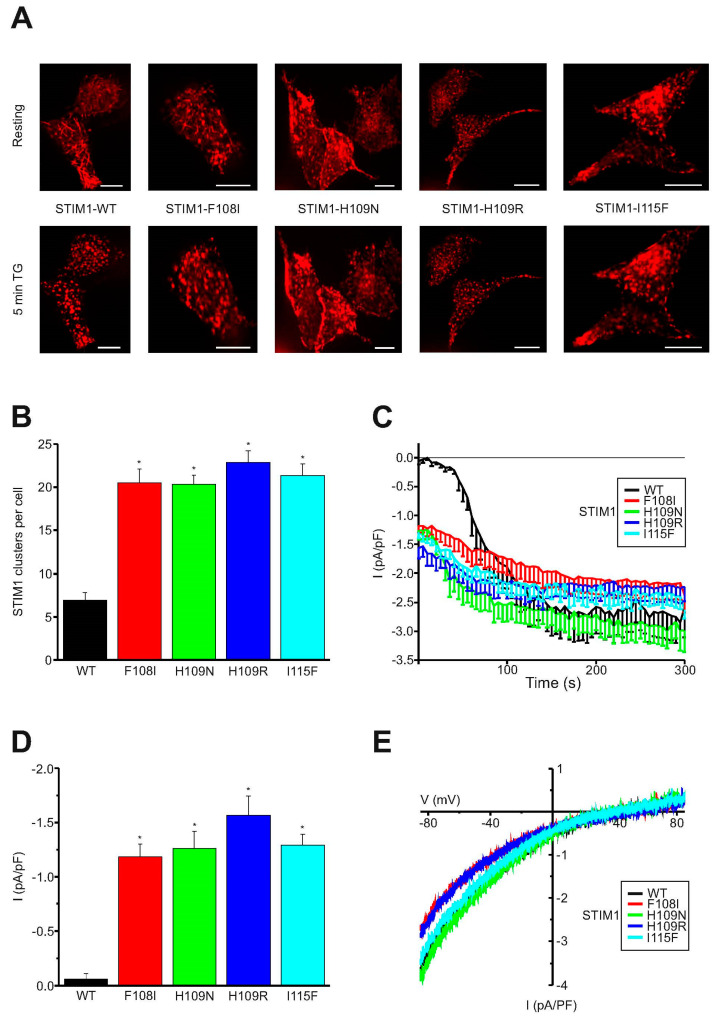
STIM1 non-canonical EF-hand mutants constitutively activate CRAC channel currents. (**A**) Representative example images of cells expressing YFP-tagged STIM1 wild type, STIM1-F108I, STIM1-H109N, STIM1-H109R, or STIM1-I115F under resting-state conditions and after treatment with 1 µM thapsigargin (TG) for 5 min. Scale bar 10 µm. (**B**) Mean value of STIM1 cluster formation per cell during resting cell conditions for wild type STIM1, STIM1-F108I, STIM1-H109N, STIM1-H109R, and STIM1-I115F (*n* = 22 to 39 cells per mutant). An asterisk (*) indicates a significant difference in the number of STIM1 clusters between STIM1 mutants and STIM1 wild type in the resting state (*t*-test: *p* < 0.01)). (**C**) Time course of whole-cell patch-clamp average currents at −86 mV from cells overexpressing STIM1 wild type, STIM1-F108I, STIM1-H109N, STIM1-H109R, and STIM1-I115F. Intracellular solution containing 20 mM EDTA passively induced store depletion. (**D**) Mean currents after whole-cell break-in (*n* = 7 to 12 cells per mutant, data were taken from three independent transfections). An asterisk (*) indicates a significant difference in whole-cell break-in currents between STIM1 mutants and STIM1 wild type in the resting state (*t*-test: *p* < 0.01). (**E**) Representative current/voltage relationships according to whole-cell patch-clamp experiments with cells overexpressing STIM1 wild type, STIM1-F108I, STIM1-H109N, STIM1-H109R, and STIM1-I115F, respectively.

**Figure 4 ijms-21-04410-f004:**
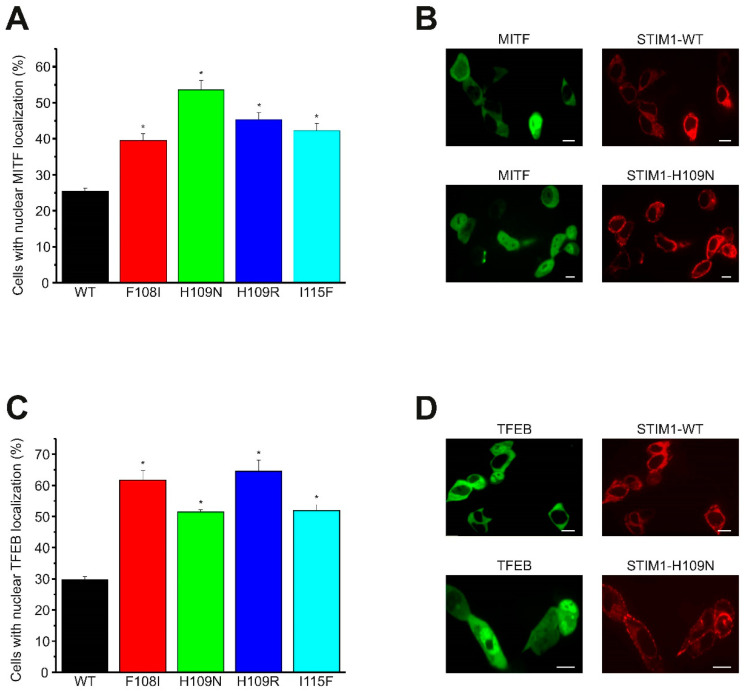
Nuclear trans-localization of microphthalmia-associated transcription factor (MITF)/TFE transcription factor family induced by heterologous expression of non-canonical EF-hand mutants. (**A**,**C**) Statistical analysis provided the percentages of cells with nuclear localization of (**A**) CFP-tagged MITF or (**C**) CFP-tagged transcription factor EB (TFEB) upon co-expression with STIM1 wild type, STIM1-F108I, STIM1-H109N, STIM1-H109R, and STIM1-I115F, respectively. For each construct, *n* = 173 to 323 cells were measured on three different days. An asterisk (*) indicates a significantly increased number of cells with nuclear MITF or TFEB localization compared to wild type STIM1 co-expression (*t*-test: *p* < 0.01). (**B**,**D**) Representative example images of cells co-expressing (**B**) CFP-tagged MITF or (**D**) CFP-tagged TFEB and YFP-tagged STIM1 wild type, and STIM1-H109N, respectively. Scale bar 10 μm.

**Figure 5 ijms-21-04410-f005:**
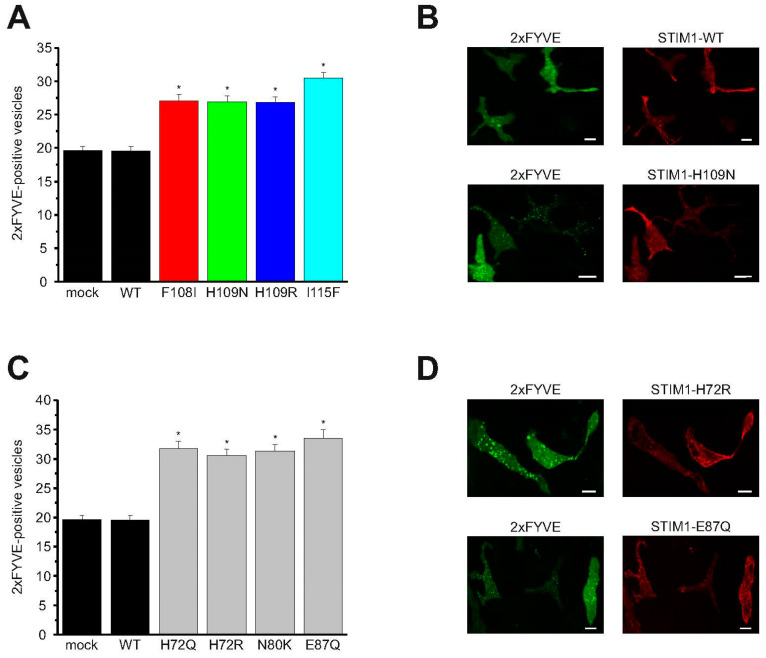
Activation of autophagic pathways by STIM1 canonical and non-canonical EF-hand mutants. (**A**) Average number of 2xFYVE positive vesicles per cell with overexpression of no STIM1 (mock), STIM1 wild type, STIM1-F108I, STIM1-H109N, STIM1-H109R, and STIM1-I115F, respectively. Each mutant was measured on three different days with *n* = 66 to 110 cells. An asterisk (*) indicates a significantly increased number of 2xFYVE positive vesicles found in cells overexpressing STIM1 mutants compared to cells overexpressing STIM1 wild type (*t*-test: *p* < 0.01). (**B**) Representative images of cells co-overexpressing lysosomal marker 2xFYVE as well as STIM1 wild type and STIM1-H109N, respectively. Scale bar 10 µm. (**C**) Average number of 2xFYVE positive vesicles per cell with overexpression of no STIM1 (mock), STIM1 wild type, STIM1-H72Q, STIM1-H72R, STIM1-N80K, and STIM1-E87Q, respectively. Each mutant was measured on three different days with *n* = 45 to 110 cells. An asterisk (*) indicates a significantly increased number of 2xFYVE positive vesicles found in cells overexpressing STIM1 mutants compared to cells overexpressing STIM1 wild type (*t*-test: *p* < 0.01). (**D**) Images representing cells co-overexpressing lysosomal marker 2xFYVE as well as STIM1-H72R and STIM-E87Q, respectively. Scale bar 10 µm.

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
