# Peer review of "Luminal STIM1 Mutants that Cause Tubular Aggregate Myopathy Promote Autophagic Processes"

_ijms, 2020, doi:10.3390/ijms21124410_

Round 1
Reviewer 1 Report
In their study, Schindl and colleagues functionally analyze STIM1 mutants (found in tubular-aggregate myopathy (TAM), cancer) that are involved in autoghagic processes. Via MD simulations and wet lab experiments they demonstrate that mutations in the non-canonical EF hand in the N-term in STIM1 contribute to altered SOCE, ICRAC and increased autophagy transcription factors MITF and TFEB and enhanced production of autophagosomes. Their work demonstrates that the underlying mechanism lies within an altered folding of the non-canonical EF hand and results in changed unfolding process upon Ca2+ depletion.
This work is of high originality and significance in the field of STIM1, Orai channels, Ca2+ signaling, MD simulation, molecular medicine, biomedical research and autophagy and will be of interest to many readers. The design of the study and the experiments (MD and live cell experiments, Ca2+ signaling and patch clamp) are well performed. The quality of presentations, scientific soundness is very good. I am rather not qualified to judge English level.
My overall recommendation is to publish this study.
However, I have some minor points:
- L31-33, Activation of this sustained SOC entry (SOCE) is provided on a molecular basis by the ER Ca2+-sensing proteins STIM1 and STIM2 and the plasma membrane (PM) Ca2+ channels Orai1, 2 and 3 [5-7]. The authors introduce STIM2 and Orai2 and Orai3, the references are only for STIM1. The first authors to characterize STIM2, Orai2 and Orai3 ICRAC were probably Parvez et al and Lis et al.
- L34-36, Resting Ca2+ concentrations are at around 800 μM, while a release of ER Ca2+, physiologically mediated by inositol-3-phosphate or ryanodine receptors, leads to a drop in ER Ca2+ concentrations to 100-400 μM. Please provide a reference for these Ca2+ concentrations
- This paper demonstrates a role for STIM1 in phagosomal maturation and may be cited, `STIM1 promotes migration, phagosomal maturation and antigen cross-presentation in dendritic cells`. Nunes-Hasler et al.
- Figure 1 C and D, I cannot find Helix 2 and 3
- Figure 3A Can preclustering be quantified?
- I cannot find the explanation for the abbreviation TAM
- L268 Space after cites is too much
- Does the rearrangement of STIM1 EF hands also effect upstream STIM1 domains such as the lately found CalM regulation site (V375, Bhardwaj et al. 2020, Ca 2+/Calmodulin Binding to STIM1 Hydrophobic Residues Facilitates Slow Ca 2+-Dependent Inactivation of the Orai1 Channel). It would be nice if the authors could discuss this.
Author Response
Reviewer 1
My overall recommendation is to publish this study.
However, I have some minor points:
L31-33, Activation of this sustained SOC entry (SOCE) is provided on a molecular basis by the ER Ca2+-sensing proteins STIM1 and STIM2 and the plasma membrane (PM) Ca2+ channels Orai1, 2 and 3 [5-7]. The authors introduce STIM2 and Orai2 and Orai3, the references are only for STIM1. The first authors to characterize STIM2, Orai2 and Orai3 ICRAC were probably Parvez et al and Lis et al.
We thank the reviewer for this constructive comment. We have now included the missing references:
Lis, A.; Peinelt, C.; Beck, A.; Parvez, S.; Monteilh-Zoller, M.; Fleig, A.; Penner, R., CRACM1, CRACM2, and CRACM3 are store-operated Ca2+ channels with distinct functional properties. Curr Biol 2007, 17, (9), 794-800.
Parvez, S.; Beck, A.; Peinelt, C.; Soboloff, J.; Lis, A.; Monteilh-Zoller, M.; Gill, D. L.; Fleig, A.; Penner, R., STIM2 protein mediates distinct store-dependent and store-independent modes of CRAC channel activation. FASEB J 2008, 22, (3), 752-61.
L34-36, Resting Ca2+ concentrations are at around 800 μM, while a release of ER Ca2+, physiologically mediated by inositol-3-phosphate or ryanodine receptors, leads to a drop in ER Ca2+ concentrations to 100-400 μM. Please provide a reference for these Ca2+ concentrations. This paper demonstrates a role for STIM1 in phagosomal maturation and may be cited, `STIM1 promotes migration, phagosomal maturation and antigen cross-presentation in dendritic cells`. Nunes-Hasler et al.
We have now included the missing references:
Luik, R. M.; Wang, B.; Prakriya, M.; Wu, M. M.; Lewis, R. S., Oligomerization of STIM1 couples ER calcium depletion to CRAC channel activation. Nature 2008, 454, (7203), 538-42.
Bischof, H.; Burgstaller, S.; Waldeck-Weiermair, M.; Rauter, T.; Schinagl, M.; Ramadani-Muja, J.; Graier, W. F.; Malli, R., Live-Cell Imaging of Physiologically Relevant Metal Ions Using Genetically Encoded FRET-Based Probes. Cells 2019, 8, (5), 492.
Figure 1 C and D, I cannot find Helix 2 and 3.
We have now specifically mentioned in the results section, that Helix 2 and 3 are shown in Fig. 1C, lower panel. Line 99. Additionally, we have now included a color code in figure legend 1.
Figure 3A Can preclustering be quantified?
We have now quantified clusters in STIM1-WT and mutants in our new Figure 3B.
I cannot find the explanation for the abbreviation TAM.
We have now included the abbreviation: tubular aggregate myopathy (TAM). Line 82.
L268 Space after cites is too much.
Corrected as suggested.
Does the rearrangement of STIM1 EF hands also effect upstream STIM1 domains such as the lately found CalM regulation site (V375, Bhardwaj et al. 2020, Ca 2+/Calmodulin Binding to STIM1 Hydrophobic Residues Facilitates Slow Ca 2+-Dependent Inactivation of the Orai1 Channel). It would be nice if the authors could discuss this.
As we did not yet examine any Calmodulin binding to STIM1 in this or in our other previous reports, we would like not to discuss this work as this might be too much of speculation at this point.
Reviewer 2 Report
This manuscript describes the MD simulation study of luminal STIM1, in particular focussing on the N-terminal canonical and non-canonical EF hand motifs that confer Ca2+ sensing. Comparing MD simulated structures of Ca2+-bound and Ca2+-free structures revealed unfolding of helices in these EF hand motifs. Furthermore, the effect of known pathogenic mutations, causing tubular aggregate myopathy (TAM) and which are located in the canonical and non-canonical EF hand motifs, were investigated by MD simulations and also functional assays. These functional assays included fluorescence imaging, measuring ICRAC by patch-clamp electrophysiology and studying autophagy processes in cells, using TAM-associated STIM1 mutants. The data show that the TAM-associated mutations cause partial unfolding of STIM1, which in turn leads to constitutively active CRAC channels and increased translocation of autophagy transcription factors into the nucleus, activating autophagy processes.
This study is an interesting combination of MD structural simulations and functional assays of STIM, one of the two protein key players of the SOCE process. What is more, the influence of mutations in STIM that have established pathological consequences (tubular aggregate myopathy, TAM) was investigated, which makes this study therapeutically relevant. The data show convincingly how these mutations lead to partially unfolded and therefore overactive STIM that in turn increases autophagy processes in cells. I don’t have any major issues with the manuscript, but would like to suggest the following changes in order to make it more accessible to readers outside the SOCE field:
- Please define abbreviations SAM and TAM upon first use
- Figures 1 and 2: If known, it would be illustrative to show the locations of the Ca2+ ions in the STIM1 structures.
- Are the binding affinities known for Ca2+ at the canonical and non-canonical EF hand? I thought they are different and it might be interesting to mention them in the context of the TAM mutations in the non-canonical EF hand.
- Figure 2A, TAM associated mutants: I’ve noticed that His72 (imidazole side chain) is either mutated to a Gln (primary amide side chain) or an Arg (basic guanidine side chain) at the end of helix alpha1 in the canonical EF hand; strikingly similar, in the non-canonical EF hand, His109 is either mutated to a Asn (primary amide) or an Arg (basic guanidine side chain) at the end of helix alpha3 as well. Is this coincidence? Do the MD simulations with His72 and His109 mutants give any structural clues as to how they destabilise the helix structures of alpha1 and alpha3? Do actually TAM patients have several of these mutations or is already one mutation causing symptoms?
- Figure 2 legend, line 129: “(b-d) Hydrophobic pocket residues [ADD: from panel (a)] are shown both as red spheres…”.
- In titles 2.2. and 2.3. please subscript CRAC in ICRAC
- page 5 of 16, line 148: “In the STIM1-F108 simulation, K104 and I108 exhibited the highest degree of unfolding, but substantially less than wild type in the absence of Ca2+”. Shouldn’t it be substantially more? Perhaps I’m reading Fig. 2E wrong, but the red bars for K104 and F108I in STIM1-F108I Ca2+ are bigger than the grey bars for K104 and F108 in STIM1-WT Ca2+ free?
- page 9 of 16, line 231: “…vesicles under resting cell conditions (Fig 5A, C upper images).” I am not sure which upper images the authors are referring to? Top row images of Fig 5 panels B and D?
- The puncta formation, ICRAC currents and MITF/TFE translocation studies (Figures 3 and 4) were performed with the non-canonical EF hand TAM mutants, whereas the autophagy pathways (Figure 5) were also studied with some canonical EF hand TAM mutants. I was just wondering if puncta formation, ICRAC currents and MITF/TFE translocation studies were conducted with canonical EF hand TAM mutants H72Q, H72R, N80K and E87Q previously? If yes, how do the data compare to the results for the non-canonical EF hand mutants shown in Figures 3 and 4?
Author Response
- Please define abbreviations SAM and TAM upon first use
We thank the reviewer for this constructive comment. We have now included the abbreviations: sterile alpha motif (SAM) and tubular aggregate myopathy (TAM). Line 49 and Line 82.
- Figures 1 and 2: If known, it would be illustrative to show the locations of the Ca2+ ions in the STIM1 structures.
We have now included an illustration of Ca2+ ions in Fig. 2B STIM1-WT. Ca2+ binding in MD-simulations behaves very dynamically and is therefore not included in further illustrations.
- Are the binding affinities known for Ca2+ at the canonical and non-canonical EF hand? I thought they are different and it might be interesting to mention them in the context of the TAM mutations in the non-canonical EF hand.
This is an interesting point and we believe we have carefully addressed Ca2+ binding to canonical and non-canonical EF-hands in our previous work (Schober et al. Science Signaling 2019). In the results section we have additionally stated that “These experiments are very well in line with our recent MD simulations on STIM1-F108I that loses dynamic Ca2+ interactions of the canoncial EF-hand over time”. Line 167-169.
- Figure 2A, TAM associated mutants: I’ve noticed that His72 (imidazole side chain) is either mutated to a Gln (primary amide side chain) or an Arg (basic guanidine side chain) at the end of helix alpha1 in the canonical EF hand; strikingly similar, in the non-canonical EF hand, His109 is either mutated to a Asn (primary amide) or an Arg (basic guanidine side chain) at the end of helix alpha3 as well. Is this coincidence? Do the MD simulations with His72 and His109 mutants give any structural clues as to how they destabilise the helix structures of alpha1 and alpha3? Do actually TAM patients have several of these mutations or is already one mutation causing symptoms?
Remarkably, one mutation in the non-canonical EF-hand is sufficient to cause a tubular aggregate myopathy disease (Böhm et al.). Indeed, in a previous work (Schober et al. Science Signaling 2019) we have evaluated a STIM1-H72R mutation in MD simulation that showed an unfolding of the non-canonical EF-hand. We expect the reason for this destabilizing effect is that H72 contributes to the hydrophobic pocket and mutation to an N or R makes it much more hydrophilic and hence the residue would not keep the position within the hydrophobic pocket. The same is likely true also for His109. It is indeed fascinating that this small change has such a drastic impact on the luminal STIM1 structure. After discussion of F108I mutant, we have now included in line 296 the following: “We expect similar structural effects also for other non-canonical EF-hand TAM mutants. As example, the substitution of H109 to more hydrophilic side-chains would no longer favor an orientation of the mutated arginine or asparagine in the hydrophobic pocket and hence would destabilize this pocket.”
- Figure 2 legend, line 129: “(b-d) Hydrophobic pocket residues [ADD: from panel (a)] are shown both as red spheres…”.
Corrected as suggested. Please see line 131.
- In titles 2.2. and 2.3. please subscript CRAC in ICRAC
Corrected as suggested.
- page 5 of 16, line 148: “In the STIM1-F108 simulation, K104 and I108 exhibited the highest degree of unfolding, but substantially less than wild type in the absence of Ca2+”. Shouldn’t it be substantially more? Perhaps I’m reading Fig. 2E wrong, but the red bars for K104 and F108I in STIM1-F108I Ca2+ are bigger than the grey bars for K104 and F108 in STIM1-WT Ca2+ free?
This sentence was indeed confusing we have now clarified and wrote that “In the STIM1-F108I simulation, K104 and I108 exhibited the highest degree of unfolding. The amount of unfolding of all hydrophobic residues was in general higher for the Ca2+ free STIM1 than for STIM1-F108I.”
- page 9 of 16, line 231: “…vesicles under resting cell conditions (Fig 5A, C upper images).” I am not sure which upper images the authors are referring to? Top row images of Fig 5 panels B and D?
That is indeed misleading, we have now corrected and have removed “upper images”. Please see, line 238.
- The puncta formation, ICRAC currents and MITF/TFE translocation studies (Figures 3 and 4) were performed with the non-canonical EF hand TAM mutants, whereas the autophagy pathways (Figure 5) were also studied with some canonical EF hand TAM mutants. I was just wondering if puncta formation, ICRAC currents and MITF/TFE translocation studies were conducted with canonical EF hand TAM mutants H72Q, H72R, N80K and E87Q previously? If yes, how do the data compare to the results for the non-canonical EF hand mutants shown in Figures 3 and 4?
Indeed, we have performed a similar analysis previously in Schober et al. Science Signaling 2019, and the results perfectly match that TAM mutants showed largely increased puncta formation. Please see, line 234-236.